# Development of Material Combination Model Considering Economics and Construction Efficiency for G-SEED Certification

**Byung-Ju Jeon and Byung-Soo Kim \*** 

Department of Civil Engineering, Kyungpook National University, Daegu 41566, Korea; kyung7673@knu.ac.kr
\* Correspondence: bskim65@knu.ac.kr; Tel.: +82-10-6205-5348

**Abstract:** The Korean government proposed a goal to reduce its greenhouse gas emissions by 37% compared to business-as-usual levels by 2030 and launched the Green Standard for Energy and Environmental Design (G-SEED) certification system. The certification requires meeting the required score and material selection with a secured economy and construction efficiency. However, most buildings only focus on obtaining the certification scores instead of choosing economical materials with high construction efficiency. This research focused on developing a material selection model that considers both the construction efficiency and economy of the materials and the acquisition of material and resource evaluation scores from the G-SEED certification. This research, therefore, analyzed actual data to automate the material selection and compare alternatives to using a genetic algorithm to obtain optimized alternatives. This model proposes an alternative to constructability and economy when the required score and material information is entered. When the model was applied to actual cases, the result revealed a reduction in construction costs of about 37% compared to the cost with the traditional methods. The material selection model from this research can benefit construction project owners in terms of cost reduction, designers in terms of structural design time, and constructors in terms of construction efficiency.

**Keywords:** G-SEED; material selection; constructability; economic; genetic algorithms

## 1. Introduction

*1.1. Background and Purpose*

The Korean government has proposed a goal to reduce its greenhouse gas emissions by 37% compared to business-as-usual (BAU) levels by 2030. The Ministry of Land, Infrastructure, and Transport established a greenhouse gas reduction target of 26.9% in the building sector by 2020 compared to BAU levels and set forth the First Green Building Basic Plan [1,2]. The goal of the Green Building Basic Plan is to reduce carbon dioxide emissions by activation green buildings to provide and nurture green buildings for low-carbon environments and a green lifestyle [2].

The social importance of G-SEED is rising every day. However, due to intricate verification and documentation, large agencies have emerged, with an increase in certification costs and a lack of connection between design and construction processes [3]. This has led to frequent alterations in the architecture design, which eventually takes a long time due to frequent repetitive but simple tasks. Also, some information loss occurs due to the limitation of the users' cognitive knowledge [4].

This research aims to develop a model that proposes an alternative for the building materials that are required to obtain an appropriate score, construction cost, and construction efficiency for G-SEED. Here, the study used a genetic algorithm (GA) to solve the optimization problem. The research is expected to benefit construction project owners in terms of cost reduction, designers in terms of structural design time, and constructors in terms of construction efficiency.

### 1.2. Research Scope and Method

As shown in Figure 1, the scope of this research is set as the five certification criteria for the selection of material selection '3. Material and Resources' based on seven evaluation criteria for new and non-residential buildings for G-SEED certification. '3.6 storage of recyclable resources' was not considered in the material and resources due to the lack of relevance to the use of materials. Also, the research encompasses the solution for the selection of a mixture of materials that meets the G-SEED score and satisfies the construction efficiency and economical requirements of several alternatives in the selection of materials.

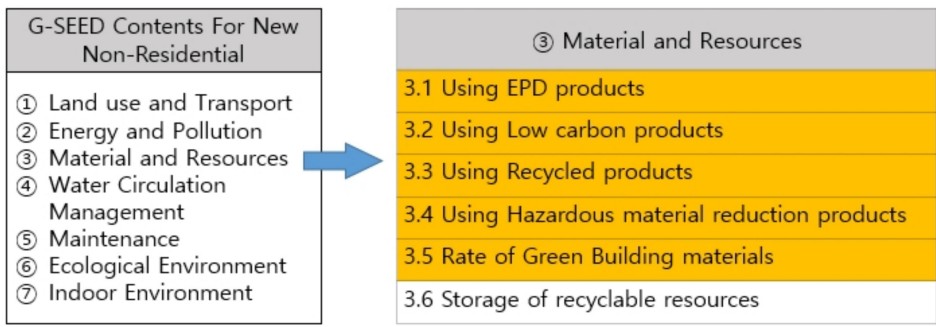

**Figure 1.** Scope of research.

Literature and theoretical reviews were conducted on the existing G-SEED buildings that were conducted for the research method, as shown in Figure 2. The review was performed to understand the concept of certification and the current status of G-SEED and to study the evaluation criteria.

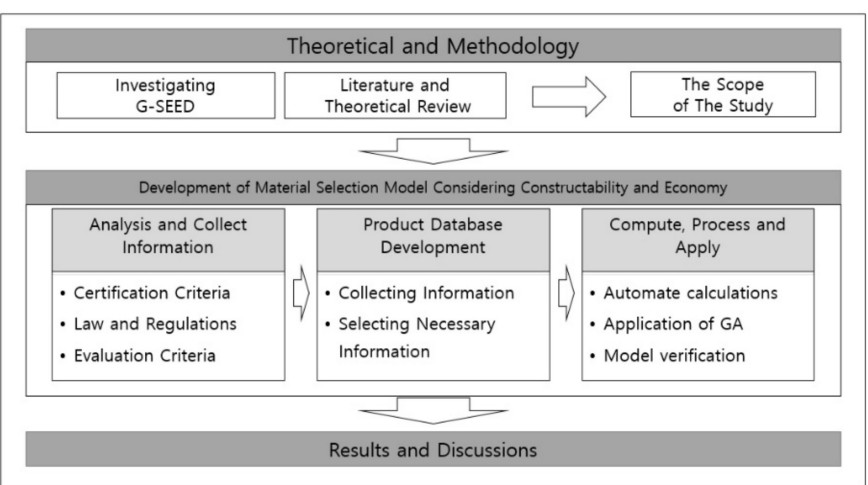

**Figure 2.** Process of research.

Second, the data on the cost of materials and eco-friendly certification information were collected and processed. Information on the status of eco-labeled products, green construction material information, and products certified via the Environmental Product Declaration (EPD) provided by the Ministry of Environment and Korean Environmental Industry and Technology Institute was provided.

Third, the evaluations studied were used to automatically calculate the scores for the certification category for each material alternative. The GA was then used to build a model that meets the construction efficiency requirements and minimizes the construction cost. Evolver (v 7.6), an optimization program, was used. The model was verified by applying quantities and target scores of existing cases.

The GA was used in this research because it exhibits excellent performance in determining the optimal solution to a complex problem. This research must solve both the construction efficiency and economic issues of many material alternatives through optimization, where the GA is the best fit that can search for a solution through the optimization of the algorithm.

### 1.3. Existing Research Review

In Korea, G-SEED-certified buildings have consistently increased since 2011. However, no existing buildings are registered, and the government does not offer sufficient incentives to encourage private sector participation in the G-SEED or raise awareness about the importance of green buildings [5]. Considerable research has been conducted to propose improvements to the current issues of the G-SEED through the study of domestic and international cases and comparing these cases to similar certification systems of other countries [1,6–9]. Also, research on construction and energy costs has been conducted [10–12]. Again, other studies have (1) investigated the development of a Green BIM Template to evaluate the G-SEED based on BIM [13]; (2) a comparative study on change in pre-certification cases was conducted before and after September 2016 [3]; and (3) apartments are divided by class, region, and number of households to analyze the change in score acquisition by type [10].

Overseas, research on LEED (Leadership in Energy and Environmental Design) and BREEAM (Building Research Establishment Assessment Method) is mainly conducted. By investigating the relationship between BREEAM and LEED items, the correlation between the LEED score and the BREEAM score was found [14].

For LEED certification, energy analysis was performed through drawings created in connection with BIM, and cost information was received from the database to calculate the cost according to the energy performance score [15]. In a similar case, that evaluated and scored the sustainability of LEED based on BIM design data [16]. Also, some studies facilitated the analysis through data exchange between GBAS (Green Building Assessment Schemes) and BIM (Building Information Modeling) [17].

The following studies have analyzed the effects that can be obtained through building certification, additional costs, and reduced maintenance costs. Some research did case studies about an office building in New York city for comparing energy efficiency between LEED-certified cases, that have been ENERGY STAR scored, with not-certified ones [18]. Besides, there was also a study that evaluated the relevance of marketing by synthesizing the LEED-certified cases and by synthesizing the certification grades and scores for each case [19]. The thesis focused on cost research is a study that analyzes the cost of each score for obtaining LEED grades, and investigates the minimum cost for certification [20]. Also, in the case of LEED certification, there is a thesis that analyzes the time it takes to recover the additional construction cost by estimating the amount of additional construction cost and energy savings [21].

As a material selection model similar to this study, there existed a local model in Vietnam and a model for the LEED score. Vietnam's model created a model that considers both the number of working days and the cost of materials to meet the minimum cost to receive a score for certification [22]. In the study of the model for selecting materials for LEED, a model was developed to derive a material proposal that satisfies the optimal material alternative for obtaining a score at a minimum cost [23,24].

### 1.4. Research Significance

In this way, many studies have been conducted through their methods on optimizing the cost of obtaining LEED scores and obtaining the least-cost materials for obtaining the scores [23,24]. In particular, in Vietnam, new factors are being considered, such as developing a model that considers labor according to materials [22]. However, researchers in Korea have so far primarily proposed improvements to the G-SEED certification system by analyzing and comparing the limitations of the current G-SEED system to those of

foreign institutions. Some have analyzed the effect of the system on the reduction of construction costs but most studies were limited to proposing improvements and the effects of reducing construction costs. In other words, research on material optimization is not only inferior to that of overseas research, but it is also an obstacle to the spread of G-SEED, as it feels like a barrier for construction companies to obtain certification.

Therefore, in this study, a genetic algorithm was used as a methodology to develop a model that can derive a material plan that satisfies the user's desired score at an optimal cost. Also, this model can consider the specificity of G-SEED, which calculates scores through the number of applied materials, this model tries to develop a model that can secure construction efficiency by supplementing it to meet this by using as few materials as possible. This is a unique feature of this study that differentiates it from models studied abroad, and it compensates for the problem that the construction efficiency may be degraded by using a large number of materials to receive a lot of G-SEED scores.

The importance of this study is that it advances the research related to optimization of the lagged G-SEED one step further and reflects the characteristics of the system to have a characteristic advantage different from other material alternative derivation models and reduces the burden on building owners by using this study. It can contribute to the spread of the green building certification system.

## 2. Theoretical Review

### 2.1. Green Standard for Energy and Environmental Design: G-SEED

The G-SEED system is designed to reduce environmental burdens, such as energy and pollutants emitted from material selection, design, and construction to maintenance and disposal through the entire cycle of the building. The goal is to encourage a pleasant environment by assessing the eco-friendliness of the building [10].

Also, the G-SEED has specified score criteria for the building class. The classes include apartments, complex buildings (residential), commercial buildings, school facilities, sales facilities, small houses, and other new and existing buildings (apartments and commercial buildings). The seven criteria include (1) land use and transportation, (2) energy and environmental pollution, (3) materials and resources, (4) hydrological cycle management, (5) maintenance, (6) the ecological environment, and (7) the indoor environment, as illustrated in Figure 1 [7]. As listed in Table 1, the scores are best, excellent, super, and normal.

**Table 1.** Score criteria for each grade of the facility.

| Division | | Best | Excellent | Super | Normal |
|---|---|---|---|---|---|
| New Building | Residential | Over 74 points | Over 66 points | Over 58 points | Over 50 points |
| | Detached House | Over 74 points | Over 66 points | Over 58 points | Over 50 points |
| | Non-Residential | Over 80 points | Over 70 points | Over 60 points | Over 50 points |
| Existing | Residential | Over 69 points | Over 61 points | Over 53 points | Over 45 points |
| | Non-Residential | Over 75 points | Over 65 points | Over 55 points | Over 45 points |
| Green Remodeling | Residential | Over 69 points | Over 61 points | Over 53 points | Over 45 points |
| | Non-Residential | Over 75 points | Over 65 points | Over 55 points | Over 45 points |

### 2.2. GA

The GA is a model of biological evolution that was first developed by John Holland in 1975. The model creates a set of parameters expressed in binaries for a fitness assessment. The solution is found within a larger context by eliminating relatively inappropriate solutions and mutating good solutions using probability rules. Because of this known advantage, the algorithm is considered to be the best solution for the optimal solution in a complex problem [4].

The basic process of a GA is displayed in Figure 3. It consists of initiation, fitness evaluation, crossover, and mutation. Initiation is the first step in creating an early-stage group that includes groups that are chosen randomly but contain possible solutions from experience. The fitness evaluation is a step in which each group undergoes a fitness function to determine which solutions are considered more fit. The solutions that exhibit a better fit are replicated, and solutions that are less fit are eliminated. During the crossover stage, these selected solutions are reunited, creating a new solution by exchanging genetic information. The mutation stage is when more than one piece of information of the selected object is randomly altered to introduce new genetic information. The new group created from this process is evaluated repeatedly until the stop condition is met to determine the most optimal solution.

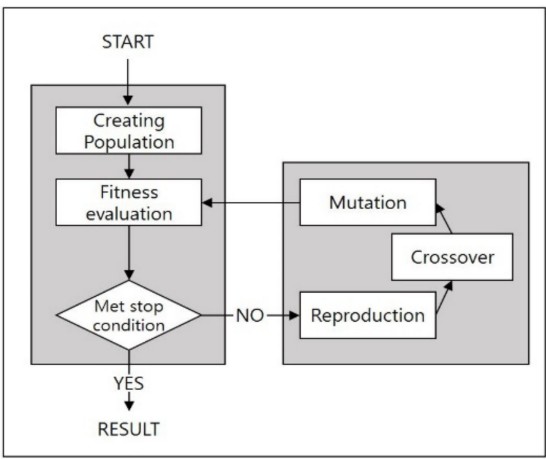

**Figure 3.** Genetic algorithm process [4].

## 3. Development of the Material Selection Model

The material selection model is illustrated in Figure 4. When the target score, basic material, and total cost are entered, new material alternatives are generated and compared to each category score and cost with the information entered. The process leverages the GA, which is known in a complex problem to determine the optimal solution. It undergoes reproduction, crossover, and mutation processes to create alternatives that are cheaper and meet higher construction efficiency requirements for the user.

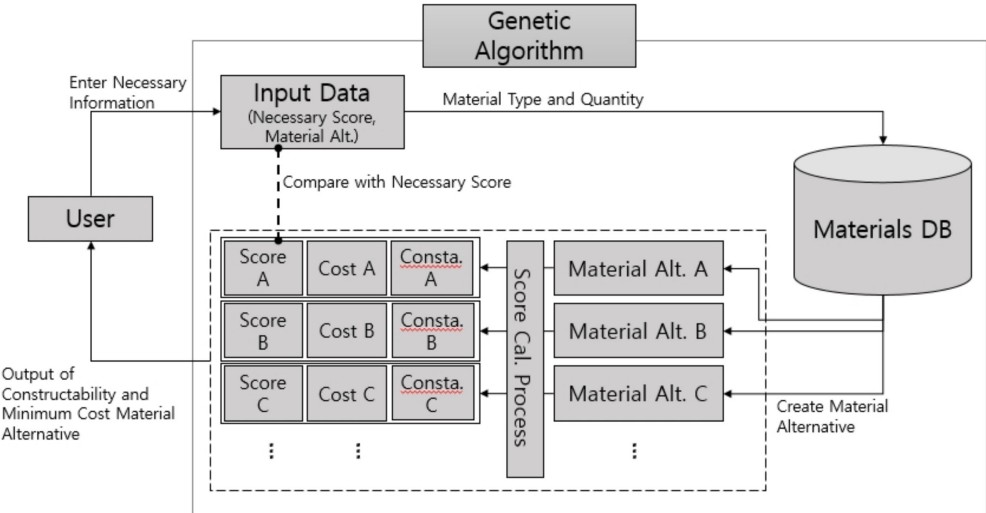

**Figure 4.** Overview of the model process.

### 3.1. Analysis of the Certification Criteria and Material Information Collection

Figure 1 presents the evaluation criteria for the new non-residential regular buildings for the G-SEED. Table 2 lists the point system for certification Articles 3.1 to 3.5 on materials and resources that fall within the scope of this research. To understand how the verification evaluation from Table 2 is applied to real buildings, an analysis was conducted on the G-SEED interpretation document, consulting documents of new non-residential regular buildings in Seoul.

**Table 2.** Evaluation criteria for materials and resources [25].

| Certification Article | Evaluation Criteria | Points |
|---|---|---|
| 3.1 Using Environmental Product Declaration (EPD) Products | When using more than six kinds of environmental declaration products from more than four kinds of main building members | 4 |
| 3.2 Using Low-Carbon Products | Seven or more low-carbon materials | 2 |
| 3.3 Using Recycling Products | 20 or more resource recycling materials | 2 |
| 3.4 Using Hazardous Material Reduction Products | 20 or more hazardous substance reduction materials | 2 |
| 3.5 Rate of Green Building Materials | When the rate of applying green building materials is more than 7% of the construction cost | 4 |

Article 3.1 includes materials with low-carbon, carbon footprint, and water footprint certifications based on interpretation and real cases. Article 3.2 includes low-carbon materials, and Article 3.3 includes resource circulation products and Green Recycled certification. Also, Article 3.4 includes hazardous materials with eco-labels.

Building a database of basic material based on the analysis of materials required for the G-SEED certification requires a database of certification information for the green building materials. Table 3 presents the results following the collection of the materials required for research, including environmental mark certification products, eco-friendly construction material information, and environmental declaration production information.

**Table 3.** Green construction material information.

| Data Name | Certification Content | Issuer |
|---|---|---|
| Environmental Mark Certification Products | Global Environmental Pollution Reduction, Local Environmental Pollution Reduction, Hazardous Substance Reduction, Recycling Effective Resources, Improving Resource Circulation | Environmental Industry and Technology Institute |
| Eco-friendly Construction Material Information | Material Unit Cost | Environmental Industry and Technology Institute |
| Environmental Declaration Production Information | Environmental Label, Water Footprint, Carbon Footprint, Low-Carbon Products | Ministry of Environment |

### 3.2. Building Material Database

The G-SEED certification material database was built because information on various unit prices of materials and certification is needed to create a new material alternative during the modeling process. The material database was built based on the green construction material information collected from Section 3.1 of this research. In the building process, as shown in Figure 5, only construction materials and equipment from environment mark certification products were extracted to estimate the unit cost in the next stage based on the green construction material information. By adding certification information, such as the environmental report, carbon footprint, and low-carbon certification, a green building material database was built with 3688 data. Table 4 is list of database

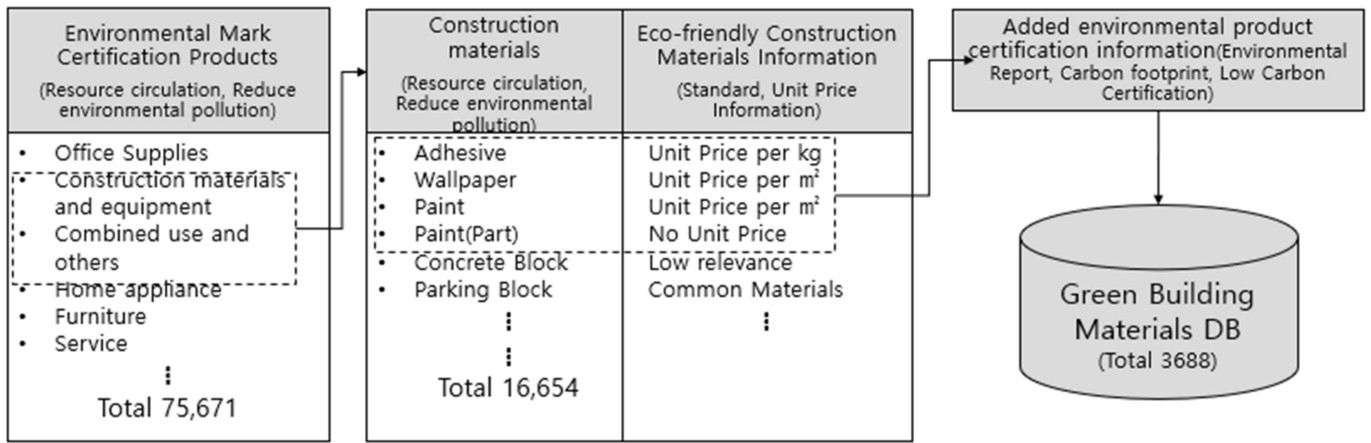

**Figure 5.** Material database building process.

**Table 4.** Integrated material database.

| Product Category | Material No. |
|---|---|
| Paint (kg) | 726 |
| Wallpaper (m$^2$) | 78 |
| Insulation board (m$^2$) | 218 |
| Waterproofing material (m$^2$) | 67 |
| Flooring (m$^2$) | 337 |
| Floor panels for heating (m$^2$) | 11 |
| Finishing material for wall and ceiling (m$^2$) | 66 |
| Finishing material for the wall (m$^2$) | 201 |
| Finishing material for the ceiling (m$^2$) | 12 |
| Windows (kg) | 882 |
| Windows (m$^2$) | 415 |
| Glue (kg) | 25 |
| OA Floor (m$^2$) | 38 |
| Sealant (L) | 41 |
| Toilet partition (m$^2$) | 287 |
| Deck (m$^2$) | 52 |
| Copper alloy (kg) | 7 |
| Clay tile (EA) | 40 |
| Clay brick (EA) | 185 |
| Total | 3688 |

For a smooth exchange among materials, the 3688 green building materials must be unified in the same unit. If the unit prices of the green construction material and the expected costs were different, the metric unit and unit prices for the product category were unified based on the product information provided by the manufacturer.

### 3.3. Class Estimation Process and Material Selection

3.3.1. Class Estimation Process

The calculation process was formed based on the G-SEED certification system to calculate the evaluation score using the built database. As shown in the database in Figure 6, each material was either labeled certified or uncertified for automated score calculations.

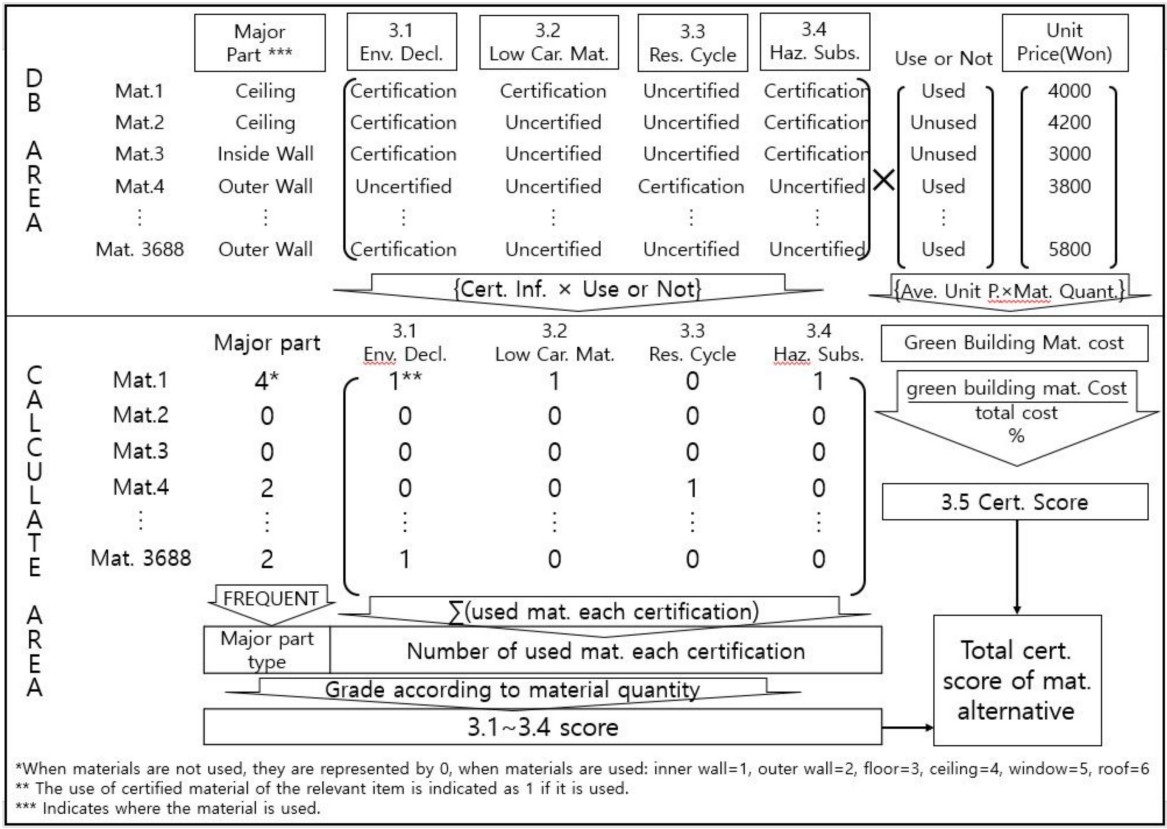

**Figure 6.** The calculation process of the material alternatives.

Low-carbon materials are considered environmentally friendly only after they have been declared environmentally friendly before and after carbon reduction. Therefore, any product of EPD, low-carbon materials, or carbon footprint certified materials are labeled as "certified" in the EPD column but are labeled "uncertified" unless they are certified by any of them. Also, several major construction materials are required as per 3.1 Using Environmental Product Declaration (EPD) Products. Building members refer to materials used to construct different parts of the building. In the database, building members are categorized into the inside wall, outer wall, floor, ceiling, window, and roof. For example, wallpaper is mostly used for building interiors, which are marked as "inside walls". Windows are marked separately because they have a separate category.

Products are marked "certified" in the certification Article '3.2 Using Low-Carbon Products' only when the material is certified as a low-carbon material. Materials are labeled "certified" in '3.3 Using Recycling Products' if they include "resource" in the certification category. Materials are labeled "certified" in '3.4 Using Hazardous Material Reduction Products', if they include keywords, such as hazardous, or regions in their certification category.

In '3.5 Application Ratio of Green Construction Materials', the materials are divided into different classes depending on the ratio use of green construction materials relative to the total material cost from the information provided. Because this research consists only of data on the green building material, '3.5 Application Ratio of Green Construction Materials' was not shown along with the other certifications. Rather, the unit price for each product category and the quantity required for each material were multiplied and then divided by the total cost.

The material database from Figure 6 is labeled certified, uncertified, used, or unused, but certified materials were annotated with 1, in the actual Excel-based material database, and uncertified materials were annotated with 0 for easy calculation. Also, for the use of

materials, used materials were noted with a value of 1, and unused materials were noted with a value of 0. For the major materials, the inside wall, outer wall, floor, ceiling, window, and roof were labeled in order from 1 to 6.

The certification score of '3. Materials and resources' of the G-SEED certification system were entered in the model following the calculation method in Figure 6. Depending on the certification of the used materials, the scores were calculated by counting the number of materials used from Articles 3.1 to 3.4. Also, the use ratio of the green building materials used to the total cost is 3.5. Then, each certification score was added to derive the total certification score.

To evaluate '3.1 Using Environmental Product Declaration (EPD) Product', the numbers of the total environment declaration products and the major building members are required. Three functions in Excel (sum, if, and frequency) were used to calculate the number of major building members.

For example, when analyzing the type of major building member of Materials 1 to 3 in Figure 6, the major building member of Material 1 was the ceiling. Because it was used for the ceiling construction, the total number of major building members is 1, the ceiling type. Material 2 was used for the ceiling but was unused, which means that it is not part of the calculation. Material 3 was used for the first time to construct the inner wall, but the unused material was excluded from the total calculation. As a result, the number of materials used for construction from Materials 1 to 3 is just 1, the ceiling type.

Excel's matrix calculation function was used to generate the number of certified materials for each category, which was then multiplied by the certification and use. For example, for the EPD column, if the product is certified as part of the environmental declaration, 1 is the output. However, it yields 0 if the material is not used or not certified. These values were added to each certification column using the sum function to calculate the number of certified materials for each material.

The number of major building members and the number of environmental declaration products is assigned to different scores using the "if" function. The grades depend on the number of products and building members for '3.1 Usage of Environment Declaration Products (EPD)' and are listed in Table 5.

**Table 5.** Evaluation criteria for '3.1. Usage of Environment Declaration Products (EPD)'.

| Division | Evaluation Criteria | Points |
|---|---|---|
| 1st Grade | When using more than nine kinds of environmental declaration products from more than four kinds of main building member * | 4 |
| 2nd Grade | When using more than seven kinds of environmental declaration products from more than three kinds of main building member | 3.2 |
| 3rd Grade | When using more than five kinds of environmental declaration products from more than two kinds of main building member | 2.4 |
| 4th Grade | When using more than three kinds of environmental declaration products from more than one kind of main building member | 1.6 |

* The main building member is a member that uses materials. It is divided into six types: inner wall, outer wall, floor, ceiling, window, and roof.

For '3.2 Using Low-Carbon Products' and '3.4 Using Hazardous Material Reduction Products', the numbers of low-carbon certified products, product circulation certification products, and hazardous substance reduction certification products are needed. Similar to EPD, each column for certification and usage was multiplied using the Excel matrix function, then added with the sum function to calculate the number of materials for each certification. Depending on the number of materials, the "if" function was used to show the points for each certification. The total material cost of the green building materials was divided by the total material to calculate the ratio of green construction materials for '3.5 Application Ratio of Green Construction Materials'. Afterward, the if function was used to derive the score for the ratio.

### 3.3.2. Designing the GA

For the GA, the optimization function from Palisade's Evolver (v. 7.6) was used to follow the process illustrated in Figure 4. The population size, crossover rate, and mutation rates are the three parameters and stop conditions required to run the GA. The three parameters influence the search for the optimization value of the GA. However, because there is no clear theory, trial and error is the only method to set up. Therefore, this research used Case C, a new non-residential building in Seoul, to set up a fixed value for the other two parameters while changing the third parameter continuously to determine the optimal parameter for this research. From this calculation, the optimization population was 30, the crossover rate was 80%, and the mutation rate was 10%. The simulation results are shown in Figure 7.

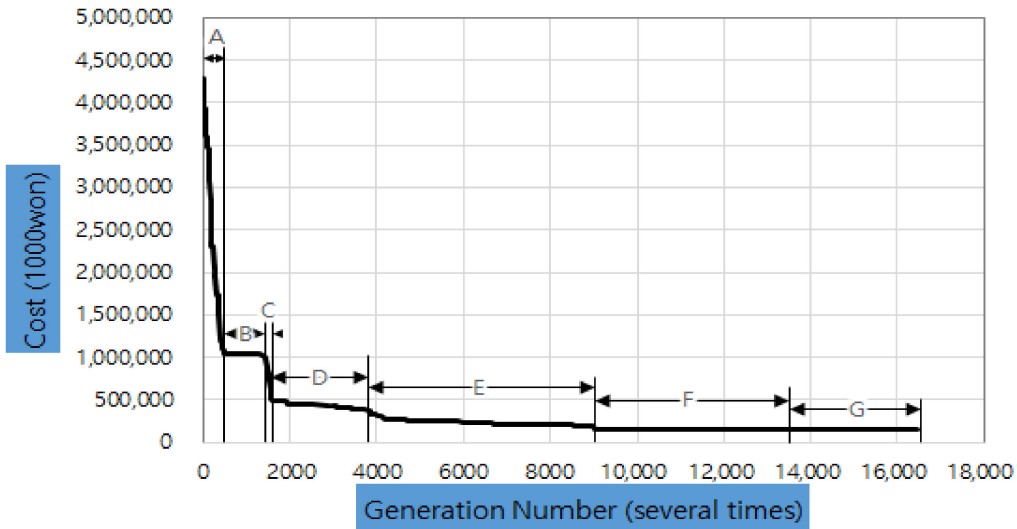

**Figure 7.** Optimization result graph for setting stop condition.

The stop condition is set by the user to prevent the repetition of the algorithm. The rate of change is chosen as the stop condition. The parameter for the change rate is the total material cost, and the stop condition for the change rate is to stop the repetition when the GA target rate does not change after a specific generation.

This research assigned time to Case A of a new non-residential building in Seoul that is likely to exhibit the greatest change and is likely to fall within the region from the lowest requirement scores of the data collected prior to setting up the stop condition for the change rate. After setting the cost for each generation for Case A as the stop condition for each generation, 16,000 generations were run to derive the optimal solution, as shown in Figure 8. For the 13,545th generation, the optimal value is 147,549 KRW. Dividing the slope of the optimal graph into seven sections revealed the greatest reduction in Section

A, followed by Sections B, C, D, E, and F and eventually stopping at Section G, which demonstrated no changes in cost, and for this case, is the optimal value.

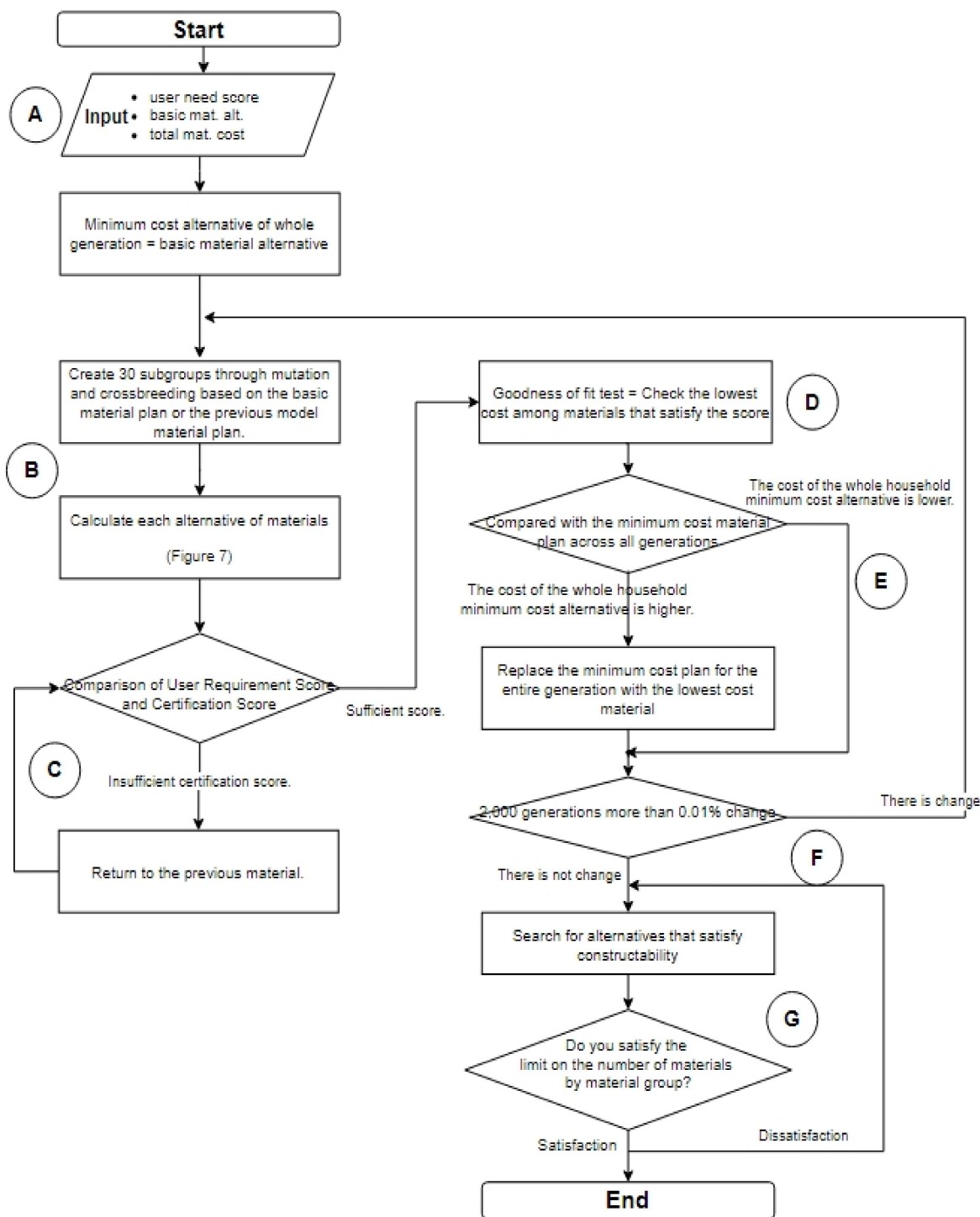

**Figure 8.** Genetic algorithm and material selection model procedure.

Section F exhibited the smallest change compared to the generation number. As shown in Table 6, the reduction rate was 2.8%, 2.4%, and 1.6% from generation numbers 9000 to 12,000, respectively. However, the change dropped to 0.06% from generations 12,000 to 13,000, with a rate of 0.001% from generations 3000 to 13,545. The change rate from generations 13,000 to 13,545 remained at 0.011%. Therefore, this research set conditions to stop

the model and produce the optimal value from Section F that demonstrated the smallest change (generations 12,000 to 13,545), where the change rate between 2000 generations is no more than 0.01%.

**Table 6.** Cost of each 1000 generations in Section F.

| Generation Number | 9000 | 10,000 | 11,000 | 12,000 | 13,000 | 13,545 |
|---|---|---|---|---|---|---|
| Initial Cost (1000 Won) | 158,225 | 153,763 | 150,024 | 147,643 | 147,550 | 147,549 |
| Reduce Amount (1000 Won) | - | 4462 | 3739 | 2381 | 93 | 1 |
| Reduction Rate (%) | - | 2.8% | 2.4% | 1.6% | 0.06% | 0.001% |

However, the alternative material produced from this calculation is designed to use many types of materials. This is shown in Table 7, where an average of 771 types of materials yielded four cases. To manage material and construction more efficiently on-site, the number of material types must be reduced as much as possible. Table 8 shows minimum material numbers per certification content and type of material by the material group. Therefore, this research employed three limitations, which are the limitations on the certification criteria, limitations on the material group, and the total limitations. The limitations for the certification and type of materials by the material group are listed in Table 9.

**Table 7.** The number of materials according to the unrestricted model.

| Division | Case A | | Case B | | Case C | | Case D | |
|---|---|---|---|---|---|---|---|---|
| | Reduction Rate * (%) | Material No. | Reduction Rate * (%) | Material No. | Reduction Rate * (%) | Material No. | Reduction Rate * (%) | Material No. |
| Restrict the number of materials per certification item | 84 | 693 | 22 | 1021 | 1 | 330 | 41 | 1,041 |

* Reduction Rate = (Material Cost in Consulting Data–Modeling Cost)/Material Cost in Consulting Data × 100.

**Table 8.** Minimum material numbers per certification content and type of material by the material group.

| Division | Certification Contents | | | |
|---|---|---|---|---|
| | 3.1 Using Environmental Product Declaration Products | 3.2 Using Low-Carbon Products | 3.3 Using Recycling Products | 3.4 Using Hazardous Material Reduction Products |
| Minimum material numbers | 9 | 9 | 25 | 25 |
| Type of material by material group | Insulation board, waterproofing material, wall finishing material, wallpaper flooring, paint, toilet divider, sealing material, glue | | | |

**Table 9.** Savings and number of materials according to the restricted model.

| | Division | Case A | | Case B | | Case C | | Case D | |
|---|---|---|---|---|---|---|---|---|---|
| | | Reduce Rate * (%) | Material No. | Reduce Rate * (%) | Material No. | Reduce Rate * (%) | Material No. | Reduce Rate * (%) | Material No. |
| (1) | Restrict the number of materials per certification item | −39 | 33 | −9 | 38 | −1 | 47 | 17 | 49 |
| (2) | Restrict the number of materials by material group | 82 | 49 | 22 | 40 | 1 | 40 | 41 | 50 |
| (3) | Restrict the total number of materials | 82 | 40 | 22 | 70 | 1 | 80 | 41 | 60 |

* Reduction Rate = (Material Cost in Consulting Data–Modeling Cost)/Material Cost in Consulting Data × 100.

Figure 8 shows the material selection process. The prerequisites shown in Case A (i.e., the user-need score, basic material alternative, and total material cost) must be entered before running the model. Here, the user-need score is used to determine if each basic material alternative meets the required score, and the materials that do not meet the score are switched to a different material that satisfies the score. To calculate the score in Article 3.5, the total material cost is needed. Because the GA runs to drive the optimal solution based on the information entered that meets the user requirements, the basic material alternative information must be entered.

The basic material plan is recorded as the minimum cost for the entire generation from the first execution. As shown in Case B, 30 groups with the same information are created randomly, where each group forms a number for each material plan and changes the usage status if the number is smaller than the change rate. These created groups undergo calculations as indicated in Figure 6. Subsequently, the scores are evaluated if they meet the user requirement scores, and the group that does not meet the score is replaced with the basic material plan.

Once all groups meet the requirements, each group undergoes a fitness test that selects the group with the highest fitness, as shown in Case D. In this research, the group with the lowest cost is considered the best fit. As shown in Case E, once the group with the lowest cost is found in the first generation, it is compared to the lowest cost for the entire generation. If the lowest cost from the first generation fits more than the cost for the entire generation, the lowest cost is replaced with the value from the first generation. After checking the stop condition (2000 generations with more than 0.01% change) shown in Case F, any change larger than 0.01% returns to Case B to proceed with the next generation. If no change occurs, it searches for alternatives that satisfy construction efficiency, as shown in Case G. Once it meets the limitations per material group for the number of materials, the final material alternative is produced.

### 3.4. Material Selection Results

Table 10 presents the results of the material selection through the material selection model. The model that does not limit the number of materials may seem appropriate in reducing the cost but uses too many materials, which poses a major challenge in material and construction management. However, the model that limits the number of materials for each material group by 10 showed similar reduction rates in materials but uses less than 50 materials. The number of green building materials seems appropriate because Table 2 shows at least 53 used green building materials to qualify for the highest grade in G-SEED. For each case, the highest score for certification is 14, and the average score for cases used in this research is 11.1, which is just below the highest score.

**Table 10.** Results of the material selection.

| Case | Real Data | | Unrestricted Model | | | | Restricted Model by Material Group | | | |
|---|---|---|---|---|---|---|---|---|---|---|
| | Material Cost (1000 won) (A) | Required Point | Material Cost (1000 won) (B) | Reduction Rate (A − B)/A × 100 | Earned Point | Using Material Number | Material Cost (1000 won) (C) | Reduction Rate (A − C)/A × 100 | Earned Point | Using Material Number |
| A | 780,223 | 8 | 124,217 | 84% | 8.4 | 693 | 141,093 | 82% | 8 | 49 |
| B | 879,497 | 12 | 686,053 | 22% | 12.4 | 1021 | 686,040 | 22% | 12.4 | 40 |
| C | 676,078 | 10.4 | 671,047 | 1% | 10.4 | 330 | 671,155 | 1% | 10.4 | 40 |
| D | 1,230,458 | 14 | 731,825 | 41% | 14 | 1041 | 731,837 | 41% | 14 | 50 |
| Total | 3,566,256 | | | | | | 2,230,125 | 37.5% | | |

## 4. Conclusions

Choosing materials to achieve eco-friendly building certifications such as LEED and BREAM is not easy. In particular, in the case of G-SEED, unlike other certification systems, points are given according to the number of applied materials. In addition, certification items must be considered all at once in the material selection process. To solve this problem, this study presented a model that satisfies the score demanded by users, optimizes cost, and creates material alternatives that consider construction effectiveness.

In this study, a database was constructed by selecting 3688 data applicable to G-SEED based on data from related organizations. Using this database, we have developed a model that automatically presents a list of recommended materials using genetic algorithms when the number of materials and the score requested by the user are entered. To verify the developed model, we performed a simulation on a G-SEED-certified building located in Seoul. As a result of the verification, the construction effectiveness targeted in this study was not satisfied.

To compensate for this, we tried to improve the model by considering the limitations on the type and number of each certification item and material. In each condition, we were able to find the number of materials that would yield the optimal cost. By comparing the cost and the number of materials, and adopting a limiting method for each material group that shows the best result, the number of materials in each material group was found to be 10 or less as optimal. As a result of applying the model developed in this study, it was found that it is possible to propose a material plan that can reduce the cost by an average of 37.5% compared to the previous one.

The challenge of this study was that it was difficult to apply the existing model due to the nature of G-SEED, which scores according to the number of materials in this way. As the number of materials increases, construction effectiveness deteriorates, and the user can obtain the result of securing the desired material combination and certification score at the optimum cost. Through this, it is expected that it will increase access to certification and provide benefits for building owners, reducing construction costs, reducing design time for designers, and considering construction effectiveness for builders. The model's algorithm is expected to be applicable to other green building certifications such as LEED.

**Author Contributions:** Conceptualization, B.-J.J. and B.-S.K.; methodology, B.-J.J.; software, B.-J.J.; validation, B.-J.J.; formal analysis, B.-J.J.; investigation, B.-J.J.; resources, B.-J.J.; data curation, B.-J.J.; writing—original draft preparation, B.-J.J.; writing—review and editing, B.-S.K.; visualization, B.-J.J.; supervision, B.-S.K.; project administration, B.-S.K.; funding acquisition, B.-S.K. All authors have read and agreed to the published version of the manuscript.

**Funding:** This research was supported by the National Research Foundation of Korea (NRF) and funded by the Korea Government (MSIT), grant number: NRF-2018R1A2B6009111.

**Institutional Review Board Statement:** Not applicable because it is not intended for humans or animals.

**Informed Consent Statement:** Not applicable because it is not intended for humans or animals.

**Data Availability Statement:** Some or all data, models, or codes that support the findings of this study are available from the corresponding author upon reasonable request.

**Acknowledgments:** Not applicable.

**Conflicts of Interest:** The authors declare no conflict of interest.

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
