# Peer review of "Development of Material Combination Model Considering Economics and Construction Efficiency for G-SEED Certification"

_sustainability, doi:10.3390/su13063535_

Round 1
Reviewer 1 Report
Dear Authors,
Thank you for the interesting manuscript with the title "Development of Material Combination Model Considering Economics and Construct ability for G-SEED Certification". Several comments for the improvement of the paper please find follow:
- Construct ability parameters are not presented in the manuscript. The manuscript title, abstract, text and etc. must be rewritten.
- The literature review and reference list of the selected problem the topic must be present newly literature source from the 2020-2021 year;
- The material art can be entitled in the presented analysis. The presented material with a material combination can be detailed in the manuscript.
Reviewer
Reviewer 2 Report
The article looks like an earlier version that needs to be greatly improved for publication.
The main body of the material analysis study is very good, but there is a lack of further argumentation of the work done.
The state of the art section is very poor. It vaguely discusses G-SEED certification and does not compare with other work in countries other than Korea.
The bibliography section is very short. A scientific article of this nature cannot be supported by only 15 references.
I think that the article is divided into many points which does not help to explain the methodology of the work. The authors should follow the guidelines of a scientific article.
The conclusions section is very basic and does not explain the reality of the G-SEED analysis. The conclusions section should be extensively justified and improved.
In conclusion, authors should arrange their work according to a standardised scientific article format, introduce the work with valid references and emphasise the conclusions of the study.
I encourage the authors to continue their work as it is an article with a lot of potential that needs to be improved.
Reviewer 3 Report
In this paper, a material selection model is proposed for construction projects for time and cost optimization. Overall, the manuscript is fine and the work is done adequately. However, for a manuscript to be scientifically appropriate and acceptable for publication, it must have novelty and usefulness from scientific perfective. The contribution to existing knowledge base is little in this paper. Mere incremental improvement in existing model does not seem much. Besides, there are several other issues.
- The introduction section Is very weak. The bibliography is too short and the review of previously published findings is not properly done.
- Further, the research gaps and niche have not been identified. The authors have not been able to bring out the novel aspect of the proposed model.
- Research significance must be provided in a separate sub-section under section 1.
- There are many lexical and grammar issues. Also, “constructability” is incorrectly used as “construct ability”. It should be rectified throughout.
- The Conclusions should be revised as well.
Round 2
Reviewer 2 Report
Good job!
Reviewer 3 Report
The paper has been revised, but still it falls short of the quality of a journal article. The issue of novelty and contribution of existing knowledge-base is there, and as such the manuscript should be declined.